# Long-term neurological consequences following benzodiazepine exposure: A scoping review

**Kyla N. Shade**[1], **Alexis D. Ritvo**[2*], **Bernard Silvernail**[3], **A. J. Reid Finlayson**[4], **Jolene E. Bressi**[5], **D. E. Foster**[6], **Ian J. Martin**[5,7], **Christi Piper**[8], **Peter R. Martin**[4,9]

1 University of Colorado School of Medicine, Aurora, Colorado, United States of America, 2 Department of Psychiatry, University of Colorado School of Medicine, Aurora, Colorado, United States of America, 3 Alliance for Benzodiazepine Best Practices, Portland, Oregon, United States of America, 4 Department of Psychiatry and Behavioral Sciences, Vanderbilt University Medical Center, Nashville, Tennessee, United States of America, 5 School of Public Health, Yale University, New Haven, Connecticut, United States of America, 6 Easing Anxiety, Erie, Colorado, United States of America, 7 Together We Can, Drug & Alcohol Recovery and Education Society, Vancouver, British Columbia, Canada, 8 Strauss Health Sciences Library, University of Colorado Anschutz Medical Campus, Aurora, Colorado, United States of America, 9 Department of Pharmacology, Vanderbilt University Medical Center, Nashville, Tennessee, United States of America

* alexis.ritvo@cuanschutz.edu

## Abstract

Benzodiazepine acute withdrawal syndrome is well known, but the long-term neurological consequences of benzodiazepine exposure are much less familiar. A scoping review was conducted of electronic databases for studies that reported on patient outcomes four or more weeks after complete cessation of benzodiazepine use. Forty-six results were retrieved in total, some of which provided signals for protracted symptoms, often reported as incidental findings, and others that showed benzodiazepine discontinuation was beneficial. Some overlap occurred in the outcomes, but these two groups of studies suggest that the benefits of benzodiazepine discontinuation for many patients tended to obscure the more prolonged, severe, and sometimes debilitating symptoms that persisted for months and years in a subpopulation of patients. The prevalence or trajectory of these enduring symptoms could not be determined from these studies. Further elucidation of the potential neurotoxicity of benzodiazepines is needed to better understand protracted symptoms and their treatment. Clinicians, patients, and the healthcare system must be cognizant of the risks of benzodiazepine exposure beyond two to four weeks.

## Background

In the 1950s, to overcome the drawbacks of barbiturates, pharmaceutical companies sought to develop safer but similarly effective anxiolytic agents and sleep-promoting drugs. Meprobamate (Miltown) was released to market in 1954 by Wallace Laboratories for the same purposes as barbiturates but with lesser risk of use, and despite

**Data availability statement:** All relevant data are within the manuscript and its Supporting Information files (the 4 appendices).

**Funding:** The author(s) received no specific funding for this work.

**Competing interests:** Alexis Ritvo is contracted as the medical director for the national non-profit the Alliance for Benzodiazepine Best Practices (501(c)(3)). Bernie Silvernail is the CEO of the Alliance for Benzodiazepine Best Practices. This does not alter our adherence to PLOS ONE policies on sharing data and materials.

commercial success, it became evident that physical dependence made it difficult to withdraw from this drug without a tapering protocol, just as for barbiturates and the risks of use were not appreciably better [1]. In 1960, the first benzodiazepine, chlordiazepoxide (Librium), was released to market and became the most-prescribed drug in the United States until it was eclipsed by another benzodiazepine, diazepam (Valium) [2]. In a clinical study, dependence and withdrawal problems were associated with benzodiazepines within one year after their introduction [3]. Initial reports of dependence and sometimes severe withdrawal symptoms were overlooked because of the substantially reduced risk of overdose death [4,5].

In 1979, Senator Ted Kennedy led a Senate subcommittee hearing on the reported dangers of benzodiazepines [6]. Growing concerns with benzodiazepines led to use of antidepressants for some of the same indications for which benzodiazepines were used. An important benefit of newer antidepressants, including such selective serotonin-reuptake inhibitors (SSRIs) as fluoxetine (Prozac), was that, unlike benzodiazepines, they were thought to have no dependence liability. A confounding between anxiety and depression treatments emerged. Medications sometimes termed second-generation benzodiazepines, e.g., alprazolam (Xanax), were developed and primarily marketed as antidepressants. By 1986, alprazolam was the best-selling drug to date in the United States and was predominantly employed as an anxiolytic [7]. Alprazolam and clonazepam (Klonopin) were newer agents and more potent than diazepam [4,5]. As exposure to these drugs increased, so did reports of adverse events, especially the risks of serious acute withdrawal reactions [8]. When tolerance and a potentially severe withdrawal syndrome became recognized, benzodiazepines were recommended for short-term use only. Despite this caution, they continued to be prescribed to many patients for prolonged periods of months or years [9]. Abrupt cessation of benzodiazepines resulted in acute withdrawal symptoms of variable severity for most patients. Initially, withdrawal protocols were developed, but many patients resumed benzodiazepine use after detoxification [10]. Tapers were eventually recommended when it was recognized that short-term detoxification may not be effective for all, and protracted neurological symptoms sometimes occurred even after tapered drug discontinuation [11]. Reports emerged of prolonged problematic clinical symptoms that could persist after completion of the acute withdrawal syndrome, enduring for months or even years after benzodiazepine cessation [12].

Benzodiazepine-induced neurological dysfunction (BIND) is a relatively recent term to describe neurological consequences and symptoms persisting past the period of acute benzodiazepine withdrawal following chronic benzodiazepine use [13]. Enduring benzodiazepine withdrawal symptoms were sometimes conflated with the acute withdrawal syndrome and/or a return of the symptoms for which the benzodiazepines were prescribed. However, earlier literature had described a separate and distinct condition of "prolonged neurological dysfunction [14,15]." To the best of our knowledge, the first report of enduring neurological dysfunction associated with benzodiazepine use was published in 1984 by Heather Ashton [16]. This detailed case series reported that all 12 patients developed new symptoms after benzodiazepine cessation that lasted at least six months after drug discontinuation [16].

These symptoms included perceptual distortion, depersonalization, and paresthesia. Ashton later reported on a cohort of 50 patients [14]. At 10 months to 3.5 years after discontinuation of benzodiazepine treatment, approximately 30% were still experiencing these symptoms. Based on a review of the literature as well as clinical experience with 60 patients, it was suggested that only about one third of patients emerge symptom free following the acute benzodiazepine withdrawal syndrome [17]. Meta-analyses of cognitive effects following withdrawal reported problems with recent memory, processing speed, visuo-construction, divided attention, working memory, and sustained attention [17,18].

This area of research was neglected for decades, because symptoms after discontinuation were considered to be a return of the problems for which the medication was initially prescribed. Nevertheless, reports of prolonged symptoms continued to confront physicians in clinical practice. In the largest survey to date of individuals who reported taking benzodiazepines long term, approximately two-thirds of 1,200 active and former benzodiazepine users reported *de novo* symptoms that commenced early in the course of benzodiazepine therapy or upon its cessation [13]. More than half of respondents' reported symptoms began early in the treatment and lasted for more than one year after discontinuation. Among the most frequently reported symptoms were: low energy, difficulty focusing, memory loss, nervousness or anxiety, and sleep disturbances. Many respondents stated they had multiple symptoms, with over 40% reporting more than 17 symptoms lasting for over a year. The majority of respondents also reported adverse life consequences, such as the loss of home or business or failed relationships [13]. There is little overlap between these *de novo* symptoms and the symptoms of acute withdrawal, and their persistence and manifestations suggest that they must be considered distinct from pharmacologic withdrawal.

The acute withdrawal syndrome following cessation of chronic benzodiazepine exposure has been well described [19–21]. The incidence of acute withdrawal is dose- and duration-dependent, ranging from approximately 30% to 100% [22]. A more prolonged condition had been observed decades ago, which typically can be identified after the acute withdrawal syndrome had abated, although nomenclature was inconsistent [23]. This survey included "life consequences," which are not symptoms in the medical sense but include issues, such as marital and family problems, financial loss, and loss of home or business.

The aim of this scoping review is to examine existing evidence in the clinical and scientific literature regarding the *de novo* neurological consequences and symptoms persisting beyond the period of acute benzodiazepine withdrawal, a condition for which the term BIND was recently coined. Four research questions emerged to help understand the limitations of current evidence surrounding neurological changes following chronic benzodiazepine use. The scoping review sought to better define the acute withdrawal period, differentiate it from a prolonged withdrawal, identify signals of neurologic symptoms persisting four or more weeks after complete drug discontinuation, and identify and potentially quantify of incidence and prevalence.

There are strengths and limitations to this work. To the best of the authors' knowledge, this is the first such scoping review ever undertaken and a great deal of literature was considered with few delimiters. No randomized clinical trials were specifically designed to look for enduring symptoms following complete benzodiazepine cessation, but many studies reported prolonged symptoms. The fact that these were signals rather than primary outcomes and were often reported incidentally rather than as specific study endpoints is a weakness.

## Methods

A scoping review that allows for a broad range of studies of variable quality is most appropriate, since the search is severely confounded due to inconsistent terminology used in the existing literature. Given that symptoms rely heavily on patient reports as well as limited available data, a scoping review was also an ideal choice for investigation.

A preliminary literature search was conducted on August 9, 2023 to determine if scoping or systematic reviews had previously been conducted. Most reviews on this topic pertain to general use, dependence, treatment of alcohol withdrawal, and acute benzodiazepine withdrawal syndrome rather than enduring symptoms. Addiction-related terminology has been

modified with respect to benzodiazepines and the term "physical dependence" has been replaced by the more precise term of "physiologic dependence." However, the older literature often contains such words. The last review on the topic of enduring symptoms was conducted by Ashton in 1991, but she did not use a systematic methodology [15]. A gap in literature was observed, with initial work done in the 1980s and 1990s with very little done around the turn of the millennium.

A protocol was developed, which included experimental studies, observational studies, case studies and case series, and clinical trials, while clinical trial registrations, commentaries, editorials, correspondence, reviews, conference abstracts, and animal studies were excluded [24]. Clinical trial registrations were excluded due to a current lull in research. Grey literature was excluded due to an already low-quality body of evidence. The complete protocol is attached as Appendix 1.

The growing understanding of benzodiazepine use, withdrawal, and prolonged symptoms has resulted in changes in terminology about this condition. Keywords in wide use today, such as "protracted withdrawal," "neuroadaptation," and even "neurocognitive symptoms" were rare in the literature. The complete list of search terms appears in Appendix 2. Since the goal was to find symptoms or adverse effects that were present four or more weeks after benzodiazepine cessation, a new search strategy had to be developed. This would include articles on pathophysiology, signs and symptoms, clinical management, nomenclature, and diagnostic criteria.

Inclusion criteria allowed for literature of any geographic origin of the article or income level of the nation. Articles not published in a peer-reviewed journal or platform or not in English were excluded. All care settings were included, such as rural, outpatients, in-hospital, and so on.

Patients were ≥ 18 years of age and may have taken benzodiazepines chronically, long term, or consistently for a period of > 2 weeks and who had discontinued benzodiazepines for a period of > 4 weeks. The three terms: "chronically," "long term," and "consistently" were used as key words and their definitions have considerable overlap. Studies could include patients prescribed benzodiazepines for any number of diagnoses with the exclusion of a primary psychotic illness, including schizophrenia, or bipolar disorder. There were no specific inclusion or exclusion criteria related to type or dose of benzodiazepine. Since protracted withdrawal identifies symptoms not present prior to benzodiazepine use, all comorbidities and coexisting conditions were considered when information was available.

The scoping review was conducted following JBI evidence synthesis methodology and the PRISMA ScR guidelines [25,26]. A comprehensive literature search was designed and performed by a medical librarian (CP) for the concepts of: benzodiazepines, specifically Alprazolam, Diazepam, Lorazepam, Clonazepam, and Nordazepam; and adverse effects of drug tolerance or dependency, or substance withdrawal. The search was initially run in August 2023 and updated in December 2024.

Relevant publications were identified by searching the following databases with a combination of standardized index terms and keywords: Ovid MEDLINE(R) ALL (1946 to December 6, 2024), Embase (via Elsevier, Embase.com, 1947 to present), and APA PsycInfo (via Ovid, 1806 to December 2024 Week 1). Searches excluded the concepts of seizures, epilepsy, and alcohol withdrawal. Searches were limited to human studies and English language articles. When possible, clinical trials, conference abstracts, and editorials/comments were excluded in the search. All results were exported to EndNote 21 for removal of duplicates. See Supplementary Materials for complete search strategies. Unique results were imported into Covidence Systematic Review software (Veritas Health Innovation, Melbourne, Australia) for screening and selection against inclusion criteria.

Prior to initiation of title and abstract screening a calibrating exercise was conducted with the first 50 publications randomly pulled from the search to increase internal validity of the review process. Disagreements were resolved via discussion and any needed clarification or updates to the inclusion criteria were made at that time. All reviewers participated in the original training or completed an independent asynchronous review of the material with the original recorded session for reference prior to beginning title and abstract screening. Conflicts during screening of the titles and abstracts were resolved by group discussion amongst several available reviewers in groups of at least three people. Reviewers

independently and sequentially evaluated titles and abstracts in pairs. Following completion of the abstract screening, full text screening took place. Conflict resolution was performed by reviewers KS or AR.

Prior to initiation of data extraction, a second calibration exercise was conducted to increase internal validity and update the data extraction template as necessary. The calibrating exercise was performed with an initial set of 4 different publications chosen from the included lists of publications following full text review. A group of four reviewers; DF, AR, BS, and RF piloted the data extraction template with any relevant changes made at that time. Following completion of data extraction while reconciling data categories, another data extraction category was added (significant outcomes after more than 4 weeks post-discontinuation), and extraction re-performed for this category. The extracted variables were the following: author, year published, publication source, study identifiers, title, intended audience (geographic practice community), country in which the study was conducted, study objective, study design, study intervention duration, post intervention follow up time, study funding sources, possible conflicts of interest for study authors, age, gender, ethnicity, method recruitment of participants, total number of participants, type of benzodiazepine, duration of treatment with benzodiazepine, tapering approach/protocol, time after discontinuation, significant outcomes after > 4 weeks post discontinuation, new onset symptoms, life consequences, other medications or interventions, reported instruments. A descriptive analysis of extracted data was performed, which identified article type and summarized relevant data as it relates to our research question to look for signals of BIND. All conflicts following data extraction were resolved by AR or KS. The PRISMA checklist is attached as Appendix 3.

## Results

A total of 14,097 publications were retrieved, of which 46 were included after eligibility assessment. No additional references were added after performing a back search by screening the references of included papers. See Fig 1.

Of the studies retrieved in the search, 41% reported on randomized clinical trials. The bulk of the literature was published prior to 1996 and there were comparatively few results after 2006. See Fig 2.

Four main themes emerged from the 46 studies. Nearly all of the studies described the acute withdrawal phase and many recognized that even after benzodiazepine cessation, some patients resumed the use of benzodiazepines. Second, there was some evidence that many new and varied symptoms indicative of neurologic dysfunction emerged during the taper or discontinuation that persisted four or more weeks after benzodiazepine cessation. Third, several studies showed beneficial effects attributable to stopping benzodiazepine use four or more weeks post-discontinuation. Finally, protracted withdrawal did not affect all persons, but there was no clear evidence of potential risk or protective factors for protracted symptoms following benzodiazepine discontinuation. The number and types of studies that supported the second and third observations are shown in Fig 3.

The literature is summarized in Tables 1–3. Since studies did not specifically evaluate protracted symptoms following benzodiazepine withdrawal, those findings relevant to our research question were suggestive signals rather than specific study endpoints. Studies reporting the strongest signals for symptoms after four or more weeks after benzodiazepine discontinuation appear in Table 1, while studies that demonstrate notable improvement upon benzodiazepine discontinuation appear in Table 2. There is some overlap between these two groups, but each study is assigned to only one table. Table 3 covers studies which demonstrated no notable results or which were disqualified due to lack of significant findings, study design, or other reasons.

The studies retrieved addressed our four research questions regarding acute withdrawal, the protracted withdrawal phase, the possibility of neurologic dysfunction, and possible methods to quantify the phenomenon of protracted withdrawal or find risk factors or protective features for it.

### The acute withdrawal phase

The population of benzodiazepine patients has been described as a "diagnostically heterogenous group," which aligns with these results [30,36,40,55,59,65,67]. The most common indications for benzodiazepine use in these studies were

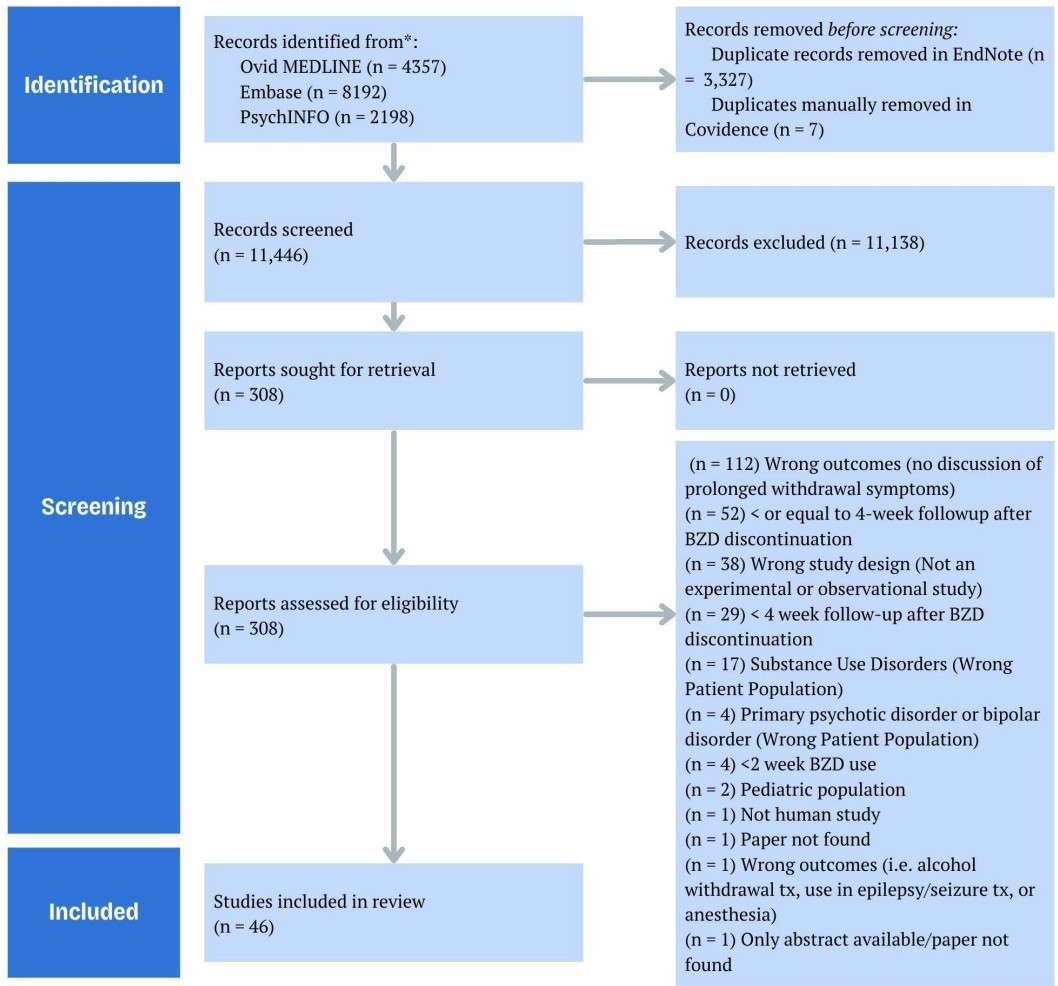

**Fig 1. PRISMA flow diagram adapted from Covidence review software.** [27] At the end of the screening, 46 articles were considered.

insomnia [30,36,40,55,59,67], generalized anxiety disorder [32,34,45,50,54,65,68], and anxiety with or without panic attacks [30,36,48,54,59,65], although studies noted the inclusion of other patients with indications such as bereavement [40] and major depressive disorder [40,65]. Enrollment criteria in these studies did not always specify indications.

Benzodiazepine acute withdrawal syndrome has been well described in the literature and is familiar to clinicians [72]. The proportion of patients who experienced acute withdrawal in our findings ranged from approximately 44% [52] to 100% [61,62]. Muscular fasciculations, mood swings, depression [53], perceptual abnormalities, dysphoria [52], trembling, nausea, vomiting, palpitations [40], headache, sweating, sensory disturbances, fear, fatigue [30], enuresis, disinhibition, aggression, impaired memory, weight loss, and ataxia [46] were reported along with the more common acute withdrawal symptoms of insomnia, anxiety, and nervousness [40]. There are established treatment protocols for benzodiazepine acute withdrawal syndrome [10,72,73].

### A protracted withdrawal phase

Following the acute withdrawal period, the emergence of *de novo* symptoms, including akathisia, pelvic pain, tinnitus, suicidal ideation, reduced appetite, weight loss, perceptual disturbances, paresthesia, depersonalization, derealization,

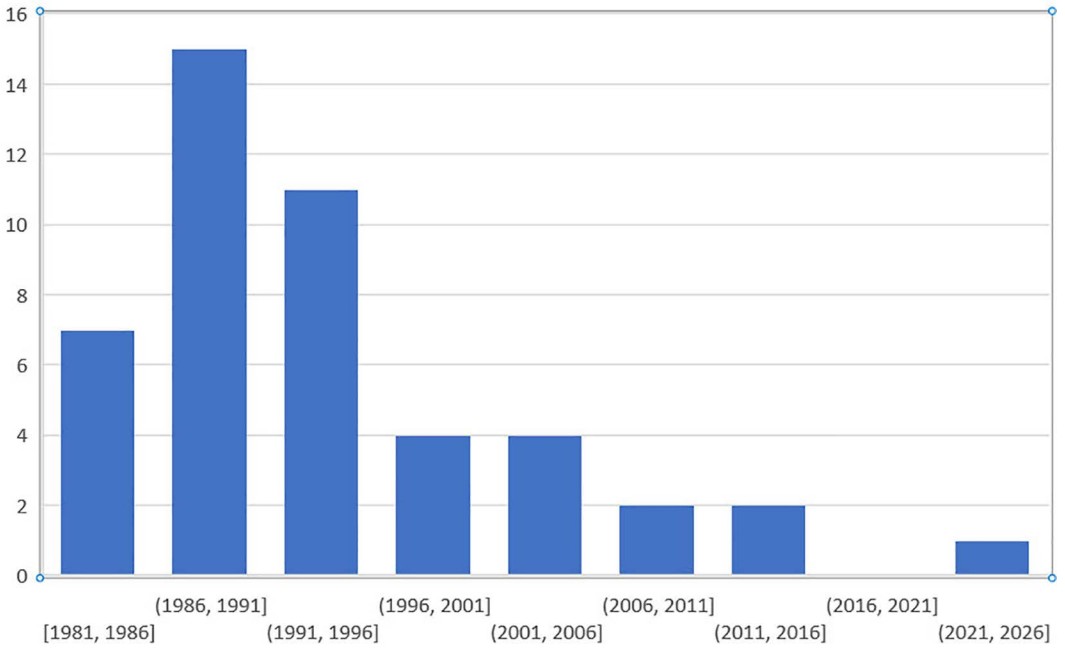

**Fig 2. Bibliometric data on study results, showing most results from the scoping review are 20 or more years old.** There is a considerable dearth of results in the period from 2006 to present.

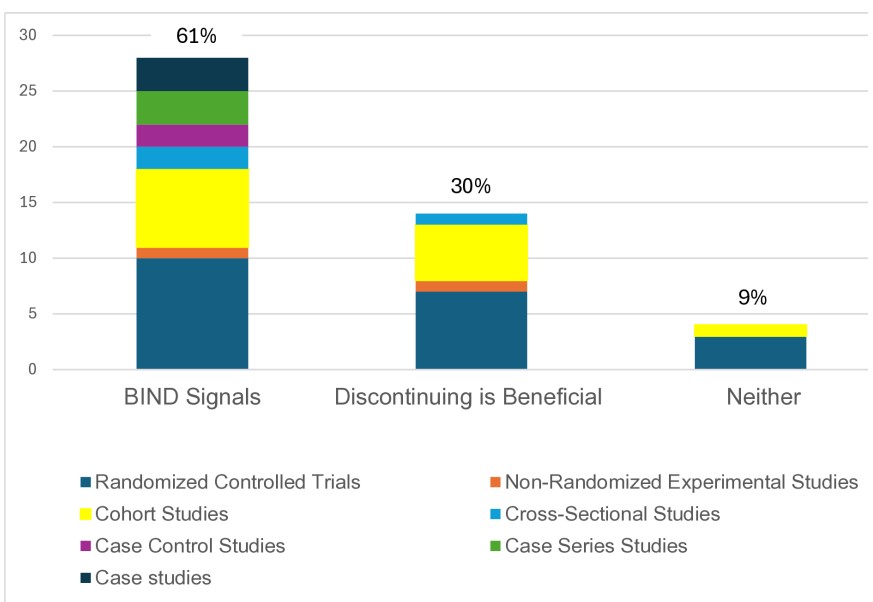

**Fig 3. The types and study designs of retrieved results, grouped by those that provided signals of benzodiazepine-induced neurological dysfunction (BIND), those that offered evidence that benzodiazepine discontinuation conferred benefits on patients, and those with no such relevant findings.**

**Table 1. Studies retrieved in the search whose results provided notable signals of protracted symptoms that persisted at least four weeks after benzodiazepine discontinuation. Studies appear in alphabetical order by surname of first author.**

| Study | Type | Description | Time After D/C | Outcomes |
|---|---|---|---|---|
| Amirzadeh-Shams 2018 [28] | Cross-sectional study | 9 BZD users who d/c matched to 24 non-BZD pt with anxiety, 13 non-BZD pt with hysteria, and 13 healthy controls | 5-24 mo | BZD withdrawal group experienced possible cognitive dysfunction as measured by EEG evoked response tests |
| Ashton 1987 [14] | Cohort study | 50 long-term BZD pt (1–22 yr) attempting withdrawal, 46 successful | 10 mo to 3.5 yr | Long-term symptoms of paresthesia in most participants; other symptoms include aggression, hallucination, impaired memory, panic attacks, headache |
| Barker 2005 [29] | Case control study | 20 prior BZD users who had d/c ≥ 1 year prior matched to healthy controls (n = 20) and controls with anxiety (n = 20) | Mean 42 mo | Compared to control groups, prior BZD users had performance deficits in verbal memory, motor control/performance, and nonverbal memory |
| Busto 1986 [30] | RCT | 40 long-term BZD users rotated to equivalent dose of diazepam (n = 21) or placebo (n = 19) | 4, 12, 24, and 52 wk | Most common withdrawal symptoms in placebo group 4 wk post-d/c were anxiety, insomnia, fear, headache, tension, tremor, sensory disturbance, fatigue, sweating, and difficulty concentrating. 2 pt had tinnitus for 8 mo post-d/c |
| Busto 1988 [31] | Case Series | Case 1: 57 yr old M<br>Case 2: 34 yr old M<br>Case 3: 49 yr old M | 1: 12 mo<br>2: 12 mo<br>3: N/A | Case 1: GI symptoms and tinnitus emerged during taper. GI symptoms resolved 6 mo post-d/c but tinnitus had not resolved at 12 mo. Case 2: Tinnitus, neck stiffness, headache, and insomnia reported during acute withdrawal; headache and insomnia resolved at 1 mo but neck stiffness and insomnia reported 3 mo post-d/c. At 6 mo, neck stiffness resolved and tinnitus was occasional and fully resolved at 12 mo. Case 3: Incapacitating tinnitus which stopped 15 min after restarting diazepam and sometimes between doses; pt was not able to d/c BZD |
| Cassano 1996 [32] | RCT | 87 entered alpidem group and 77 completed study; 86 placebo pt entered study and 68 completed | 6 wk for placebo | 17 (24.3%) in placebo group were "markedly or severely ill" on CGI scale at 4 weeks post-d/c |
| Curran 1994 [33] | RCT | 38 agoraphobics with panic disorder who had d/c BZD therapy randomized to treatment with alprazolam or placebo and psychological care (exposure or relaxation) | 5 to 8 wk | Word recall more impaired at 5–8 wk post d/c in alprazolam group vs. placebo |
| Delle Chiaie 1995 [34] | RCT | 44 generalized anxiety pt, changing to buspirone from prior lorazepam treatment | 13 wk, 6 wk post-d/c | STAI-X1 scores 6 weeks post-d/c were significantly higher (worse) compared with scores before d/c in the placebo group (p = 0.0289), and there was also a trend toward an increase for HAM-A scores (worse, p = 0.0679) in the same group |
| Golombok 1987 [35] | Cohort study | 25 successful and 21 unsuccessful BZD d/c pt surveyed | Mean 19 mo | Among those who discontinued, continuing anxiety symptoms, insomnia, and depression at follow-up |
| Golombok 1988 [36] | Cross-sectional study | 50 current BZD pt vs.<br>61 pt who had not taken BZD<br>34 who had d/c BZD for at least 6 mo | 6 mo | Those taking high doses of BZD over long periods of time performed worse on neuropsychological tests and had worse visual-spatial ability and attention spans compared to those not taking BZD |
| Gorenstein 1994 [37] | RCT | 28 anxiety pt on long-term BZD<br>Also tested: 1 mg IV flumazenil 5 days before baseline | 3 and 10 mo | Impairment in episodic memory at 10 mo |
| Gorenstein 1995 [38] | NRES | 28 pt on low-dose diazepam<br>53 anxiety pt<br>56 healthy volunteers | 10 mo | During and after BZD d/c, the diazepam group had worse performance on motor speed, coordination, and verbal recall compared to controls |
| Higgitt 1988 [39] | RCT | 9 former BZD pt vs<br>healthy controls undergoing EEG and other neurologic tests | ≥ 3 mo | Former BZD patients had higher levels than controls of: drowsiness, muscle cramps, impaired concentrations, eye soreness, and sadness |

*(Continued)*

**Table 1.** (Continued)

| Study | Type | Description | Time After D/C | Outcomes |
|---|---|---|---|---|
| Higgitt 1990 [17] | Cohort study | 9 prior BZD users with PWS with 3 matched groups:<br>9 conversion disorder pt<br>9 anxiety disorder pt<br>9 healthy volunteers | 5-42 mo | Prior BZD users who experienced prolonged benzodiazepine withdrawal symptoms (PWS) for periods of months or years after discontinuation differed from pt with conversion or anxiety disorder on psycho-physiological measures |
| Hopkins 1982 [40] | Cohort study | 49 of 78 pts d/c long-term BZD therapy | 3 and 5 mo | Symptoms associated with withdrawal that persisted at follow up include insomnia (n = 4), trembling (n = 4), lack of energy (n = 2), weakness (n = 2), and palpitations (n = 1) |
| Keshavan 1988 [41] | Case Series | Case 1: 51 yr old F<br>Case 2: 38 yr old M<br>Case 3: 51 yr old F | 1: unspecified weeks<br>2: 2 mo<br>3: 5 mo | Case 1: Hospitalized for depression during withdrawal, suicide attempt during taper, delusions<br>Case 2: suicide attempt, depression at 1 mo post-d/c, hospitalized for depression at 2 mo<br>Case 3: Depression, sleeplessness, agitation, suicidal ideation |
| Kilic 1999 [42] | Cohort study | 31 pt total<br>Alprazolam exposure (n = 7), placebo exposure (n = 9), alprazolam relaxation (n = 8), placebo relaxation (n = 7) | Interview at 3.5 yr after d/c | Follow-up of subset of patients in Curran 1994 study<br>Impaired memory observed during alprazolam use and in weeks after d/c but did not persist at 3.5 yr |
| Modell 1997 [43] | Case Report | 41 yr old F | 14 mo | Evolving symptoms 6 wk after BZD d/c including major depressive disorder with psychotic features, persistent depressed mood, poor concentration, decreased appetite, insomnia, anhedonia, anergia, psychomotor retardation, paranoid ideation, sensory disturbances. Pt had 25 yr history of IBS which recurred at 14 mo post-d/c, and she resumed BZD |
| Morton 1995 [44] | RCT | 12 buspirone vs 12 controls | 4 wk | CGI and HAM-A worsening from wk 3 of 6 wk taper to 3 wk post-d/c; improved somewhat but still above baseline in wk 4 post-d/c.<br>BWP same as HAM-A except last wk remained almost as bad as prior wk |
| Murphy 1989 [45] | RCT | 40 generalized anxiety pt taking BZD randomized to:<br>Diazepam late withdrawal, n = 10<br>Diazepam early withdrawal, n = 10<br>Buspirone late withdrawal, n = 10<br>Buspirone early withdrawal, n = 10 | 8 wk | Diazepam early withdrawal group had increased symptoms in BSA mean anxiety score between 2 and 4 wk post-d/c, followed by reduction for the next 2 wk. Mean CPRS score decreased [worsened] slightly for diazepam early withdrawal between 2 and 4 wk post-d/c and decreased more sharply for the next 2 wk |
| O'Sullivan 1996 [46] | RCT | 154 BZD pt enrolled, half treated with alprazolam for 8 wk. 124 entered withdrawal phase, 107 pt in final analysis including 9 pt who dropped out late in taper | 6 mo | BZD pt deteriorated on HAM-A and HAM-D and phobia scales for up to 6 mo post-taper. At 6 mo post-taper, CGI scores were significantly better for placebo than BZD pt (58% vs. 33%) |
| Olajide 1984 [47] | Case Series | Case 1: 61 yr old F<br>Case 2: 28 yr old M<br>Case 3: 49 yr old F<br>Case 4: 41 yr old F | 1: 10 wk<br>2: 2 mo<br>3: 7 mo<br>4: 3 mo | Case 1: No symptoms at 6 mo<br>Case 2: 4 wk after BZD d/c, symptoms of depression, social withdrawal, reduced libido<br>Case 3: Upon BZD d/c symptoms of insomnia, vomiting, muscle tension, dry heat all over body, depression, suicidal ideation. Prescribed amitriptyline and symptoms abated at 2 mo and resolved at 7 mo.<br>Case 4: After abrupt d/c suicidal ideation at 5 wk post-d/c, uncontrollable crying, social withdrawal, depressed, hospitalized as psychiatric emergency. |
| Petursson 1983 [48] | Case control study | 22 prior long-term BZD users vs. two control groups, main (n = 16) and abbreviated (n = 14) | 4-6 wk | BZD patients had ongoing deficits in sensory and fine motor skills vs controls |

*(Continued)*

**Table 1.** (Continued)

| Study | Type | Description | Time After D/C | Outcomes |
|---|---|---|---|---|
| Schweizer 1991 [49] | Cohort study | 40 BZD users who had difficulty with d/c were randomized to receive carbamazepine 200 or 800 mg/d or placebo to aid in 25% per wk taper | 5 and 12 wk | At 5 wk, carbamazepine groups had greater reduction on *Patient Withdrawal Checklist* than placebo but similar results on the *Physician Withdrawal Checklist*. At 12 wk 32% of carbamazepine and 24% of placebo pt were prescribed antidepressants; of this group, 7/11 pt had major depressive disorder and 4/11 had panic disorder |
| Silvernail 2022 [50] | Case Report | 55 yr old F | 6 yr | Variable symptoms of dizziness, generalized pain, muscle stiffness, fatigue, paresthesia, tinnitus, exercise intolerance, and hyperacusis along with life consequences |
| Tata 1994 [51] | Cohort study | 21 pt completed inpatient program and were BZD free at 6 mo matched to 21 controls | 6 mo | Impaired cognitive function, memory, psychomotor functions, delayed recall, and conceptual tracking upon BZD d/c with little improvement at 6 mo |
| Tyrer 1983 [52] | RCT | 36 diazepam out-pt to taper over 3 mo to d/c, early vs. late withdrawal | 6 mo | 3 pt became more severely ill during 6 mo follow-up and depressive illness developed in 2 of them. CPRS scores highest in early and late groups at 10 wk, decreasing by 26 wk to a value lower than baseline |
| Wilbur 1983 [53] | Case Report | 34 yr old M | 2 mo | After 2 mo of oxazepam abrupt cessation created severe anxiety and panic attacks; pt was restarted and tapered; 2 mo post d/c he had muscular pain |

BSA, Brief Scale of Anxiety; BZD, benzodiazepines; d/c, discontinuation; CGI, Clinical Global Impressions scale; CPRS, Comprehensive Psychopathological Rating Scale; EEG, electroencephalogram; F, female; GI, gastrointestinal; HAM-A, Hamilton Anxiety Rating scale; IBS, irritable bowel syndrome; M, male; mg, milligrams; mo, months; N/A, not applicable; NRES, nonrandomized experimental study; post-D/C, post-discontinuation; pt, patients; RCT, randomized controlled trial; STAI-X1, State-Trait Anxiety Inventory X1 scale; wk, week(s); yr, year(s)

numbness, and involuntary movements have been reported [30,31,50,52,61]. Twenty-eight of the 46 studies retrieved in the scoping review (61%) provide signals that suggest a protracted withdrawal phase. Months or years after benzodiazepine discontinuation, a subset of patients experienced mild to severe, even debilitating, symptoms unrelated to their prior condition, such as tinnitus, gastrointestinal symptoms, fatigue, and anxiety that do not meet criteria for another diagnosis [47,50,52,54.]

In a study of 50 patients, some patients reported symptoms ≥ 1 year after benzodiazepine discontinuation, and, paradoxically, some underlying symptoms, such as irritable bowel syndrome, resolved after benzodiazepine discontinuation [14]. A randomized clinical trial of 40 long-term benzodiazepine users tapering to discontinuation experienced symptoms of panic, anxiety, and depression to the point that 28% required antidepressant therapy at 12 weeks after benzodiazepine discontinuation, which investigators speculated may reflect *de novo* symptoms.[49] Higgitt and colleagues matched nine protracted withdrawal syndrome (PWS) patients to nine anxiety patients not taking any drugs, 13 conversion hysteria patients, and nine healthy controls. Based on psychophysiologic metrics, the PWS group was most similar to healthy controls, suggesting that protracted withdrawal is not an affective disorder, but rather an iatrogenic disorder related to benzodiazepine exposure [17].

## Evidence of neurologic dysfunction

While the majority of studies showed a range of long-term symptoms that may emerge after benzodiazepine discontinuation, neurologic symptoms are of particular concern because of their impact on daily living and the uncertain trajectory of neurological injury [13]. In one study of prolonged symptoms in 20 benzodiazepine users matched to controls, 100% of the benzodiazepine patients had neurologic symptoms at six months after complete drug cessation [51]. In a study that compared benzodiazepine patients who had ceased using benzodiazepines to benzodiazepine-free patients using

**Table 2. Studies retrieved in the literature search which showed that the discontinuation of benzodiazepines conferred benefits on patients. Many of these studies aggregated overall improvements. Studies appear in alphabetic order by surname of first author.**

| Study | Type | Description | Time After D/C | Outcomes |
|---|---|---|---|---|
| Ashton 1990 [54] | RCT | 23 chronic BZD users on taper randomized to buspirone 5 mg (n = 11) or placebo (n = 12) | 6 and 12 mo post-withdrawal | Those not taking BZD at 6 and 12 mo had less anxiety compared to those still taking BZD (p < 0.05) |
| Belleville 2008 [55] | RCT | 50 insomnia pt taking BZD | 1, 3, 6 mo | Those BZD free for 6 mo had less severe insomnia, fewer anxiety symptoms, more positive perception of health, and higher self-efficacy |
| Bourgeois 2014 [56] | Cohort study | 38 nursing home residents, goal to d/c long-term BZD use (mean age 84.3 yr) | 8 mo minus (7–88 d) | 25/38 successfully d/c BZD and had improved sleep quality over baseline |
| Cantopher 1990 [57] | RCT | 31 long-term BZD pt randomized to slow withdrawal (n = 16) or abrupt withdrawal with propranolol (n = 15) | Slow withdrawal 12 wk Propranolol 24 wk, 6 mo | Both groups (slow and abrupt withdrawal) improved on all scales at 14 wk and 6 mo. Scales included HAM-A, HAD, symptoms and global assessment of severity of illness. Symptoms assessed using the BWSQ and sleep diaries |
| Curran 2003 [58] | RCT | 104 long-term BZD users ≥ 65 yr randomized. Group A tapered at wk 1 of the study (n = 55) Group B tapered after 12 wk of usual dose (n = 49) Group C remained on same dose (n = 34) | A: 43 wk B: 31 wk | Groups A and B had better scores on cognitive and psychomotor tasks at 24 and 52 wk than Group C, but no difference in symptoms among the 3 groups |
| O'Connor 2004 [59] | Cross-sectional study | 31 BZD users participating in 20 wk taper with counseling to d/c | 3 mo | 12/33 pt succeeded with d/c. At 3 mo, those who succeeded had fewer net symptoms than others. BWSQ scores reported, not individual symptoms. |
| Peperkamp 1993 [60] | RCT | BZD users tapering to d/c Diazepam abrupt*: n = 28, Lorazepam abrupt*: n = 26, Lorazepam 2 weeks: n = 28, Lorazepam 4 weeks: n = 26 | Abrupt groups up to 5 wk post withdrawal | Final HAM-A and SQ scores much below the level at start of study. |
| Petursson 1981 [61] | RCT | 16 long-term BZD users tapering to d/c by placebo substitution in matched control study | 6 wk | Improvements in "psychological performance" (sic) soon after d/c (shown by DSST) |
| Rickels 1990 [62] | Cohort study | 63 patients taking BZD for at least 1 yr abruptly d/c the drug | 5 wk | At 5 wk those who d/c BZD had less anxiety and HAM-A scores returned to pre-d/c levels or below |
| Rickels 1991 [63] | NRES | 123 BZD pt, 55 of whom had tapered to d/c | 2.7 to 5.0 yr | Anxiety and depression levels were significantly lower for patients who had d/c BZD versus those taking BZD |
| Rickels 1999 [64] | Cohort study | 113 patients started taper; at 5 and 12 wk, 66 completed (of n = 96) | 5 and 12 wk | 77 had 12-wk data. Those able to taper to d/c showed various improvements, including 30% improvement on HAM-A with no change for BZD users |
| Schweizer 1990 [65] | RCT | 63 BZD users on 25%/wk taper | 5 wk | BZD-free patients had "modestly lower" levels of anxious and depressive symptoms vs. baseline |
| Tönne 1995 [66] | Cohort study | 11 BZD users ("steady state" group) matched to 9 withdrawal pt and 11 controls | 1 yr | At 1 yr, intelligence tests, the Halstead-Reitan battery of tests, and nonverbal memory tests were similar to controls for both BZD groups |
| Vikander 2010 [67] | Cohort study | 34 long-term BZD users who d/c BZD | 1 yr | Symptoms increased in severity when tapering started and increased to 4 wk post-taper, then reduced Symptoms not associated with duration of exposure, dose, or type of BZD |

BWSQ, Benzodiazepine Withdrawal Symptom Questionnaire; BZD, benzodiazepines; d, day(s); d/c, discontinuation; DSST, digital symbol substitution test; HAD, Hospital Anxiety Depression Scale; HAM-A Hamiliton Anxiety Scale; mg, milligrams; mo, month(s); pt, patients; SQ, Systemizing Quotient; RCT, randomized controlled trial; wk, week(s); yr, year(s).

**Table 3. These are studies which demonstrated no notable results in terms of protracted symptoms or improvement upon benzodiazepine discontinuation. These studies sometimes had results deemed to be invalidated by use of rescue medication or minor or ambiguous versions of primary results.**

| Study | Type | Description | Time After D/C | Outcomes |
|---|---|---|---|---|
| Hadley 2012 [68] | RCT | 106 pt undergoing 25%/wk taper from long-term BZD therapy randomized to pregabalin 300–600 mg/d (n = 56) or placebo (n = 50) | 6 23 wk | Study results compromised for this research by allowance of one-time use of rescue alprazolam (1 mg) |
| Lader 1992 [69] | Cohort study | 11 prior BZD pt with persistent symptoms were administered flumazenil (0.2 to 2.0 mg IV) for long-standing symptoms | 1 mo to 5 yr | List of 10 symptoms reduced post-d/c. No control group, did not separate the effect on symptoms of flumazenil from the effect of d/c |
| Romach 1998 [70] | RCT | 97 BZD users tapering to d/c, randomized to ondansetron 2 mg BID for symptoms (n = 47) or placebo (n = 50) | 52 wk | No significant results reported, other than proportion who d/c |
| Voshaar 2003 [71] | RCT | 180 long-term BZD users randomized to tapering (n = 73), tapering with group CBT (n = 73) or usual care (n = 34) | 2 mo | No significant results reported, other than proportion who d/c |

*Note that the first author of Voshaar 2003 sometimes appears in other databases and listings as Oude Voshaar. This particular article is indexed in the PubMed database with the first author's surname as Voshaar.*

BID, twice daily; BZD, benzodiazepines; CBT, cognitive behavioral therapy; d, day; d/c, discontinuation; EEG, electroencephalogram; IV, intravenous; mg, milligrams; mo, month(s); pt, patients; RCT, randomized controlled trial; wk, week(s).

control groups of anxiety patients and healthy controls, participants underwent a series of cognitive tests at three weeks post-discontinuation and again at 10 months. At both time points, the benzodiazepine patients had significantly worse scores than the controls [38]. In another study, electroencephalography testing showed that PWS patients had abnormal results when compared to healthy controls who had not used benzodiazepines [17].

Twenty participants undergoing a taper to discontinue benzodiazepines were divided into two study arms: one group was tested before the taper, the other after the taper started. Benzodiazepine patients were matched to healthy controls who took no drugs. These groups were tested again one year later. In the initial round of testing, benzodiazepine patients had test scores that showed impaired intelligence and impaired nonverbal memory versus controls but at one year, scores were similar to controls, suggesting that neurologic deficits may be at least partly, albeit slowly, reversible [66].

### Incidence and prevalence/risk stratification

Not all benzodiazepine users develop protracted symptoms after benzodiazepine discontinuation, but at present, there are no reliable metrics to quantify individual risk. This scoping review was unable to find studies describing incidence and prevalence data, and few studies offer information that might prove useful in identifying risk factors.

Acute withdrawal symptoms occurred in up to 100% of participants from studies found in this scoping review [52,61,62], but prolonged symptoms were less frequently reported, occurring in approximately 30% of benzodiazepine patients [17,54,69]. Many studies showed that benzodiazepine cessation conferred benefits on patients, and this beneficial effect may have obscured the emergence of prolonged symptoms.

### Discussion

The proportion of patients who experience prolonged symptoms and their characteristics has not been well described in the literature, likely because few benzodiazepine studies looked for prolonged symptoms. Persistent symptoms after benzodiazepine discontinuation had been observed decades earlier, but since these symptoms emerged unpredictably and only in a subpopulation of patients, they were never systematically studied. Bibliometric findings confirm an initial scientific interest in this enduring syndrome followed by a lull in research. Even nomenclature was confusing, since it was unclear

whether these protracted symptoms were a continuation of acute withdrawal, the unmasking of original symptoms, or an entirely new condition [17,52,69].

This scoping review found 46 studies of benzodiazepine cessation, of which 27 provided signals that certain symptoms persisted or emerged more than four weeks after complete benzodiazepine withdrawal. Some of these symptoms were severe, persisted for a year or more, and had life-altering ramifications for the affected patients.

Many studies in this scoping review found relatively high numbers of patients who dropped out of the study, could or would not discontinue benzodiazepines, or who discontinued benzodiazepines but resumed them a short time later. This suggests that for certain patients, benzodiazepine cessation can be far more challenging than generally appreciated. This difficulty in complete discontinuation of benzodiazepines may in itself suggest BIND.

These prolonged symptoms are many and varied. The most frequently used list of anxiolytic-associated symptoms is the Physician Withdrawal Checklist (PWC-20), but anecdotal reports of benzodiazepine-associated symptoms have appeared in the literature as well [22,74]. Neurologic, gastrointestinal, cutaneous, psychological, and other symptoms have been reported [15]. A persistent difficulty in studying protracted symptoms is differentiating them from unmasked symptoms that recurred once benzodiazepines were stopped.

Consistent with other published studies and clinical observations, this scoping review showed that some people can discontinue benzodiazepines even after prolonged exposure with relatively little problem [12,13,23]. In fact, the main finding of 14 studies retrieved (30.4%) was that, overall, benzodiazepine discontinuation was unproblematic and beneficial. However, when studies had mixed results, such that many patients discontinued without problem while others had persistent symptoms, the persistent symptoms were reported incidentally and not systematically studied. It may be that such enduring symptoms are under-reported.

The signals that emerged in the majority of the studies in this scoping review suggest that an iatrogenic condition of prolonged symptoms impacts a subpopulation of patients who have been prescribed benzodiazepines. The two primary results of the scoping review regarding what happens four or more weeks after benzodiazepines are discontinued may appear to conflict with each other. These results may be better contextualized as a continuum, and findings may depend on what investigators were seeking and where those things occurred on the continuum. Most studies that looked for enduring neurological dysfunction found it. Likewise, most studies that looked for evidence of eventual improvement following benzodiazepine discontinuation found it. This suggests that benzodiazepine cessation may reveal induced neurological dysfunction in a subset of patients, but overall confer long-term benefits to most patients.

While BIND may represent a major new challenge to the healthcare system, this is not to trivialize the importance of benzodiazepines for their specific indications: long-term treatment of acute movement disorders such as catatonia, stiff-person syndrome, status dystonicus and, more rarely, seizure disorders and burning mouth syndrome (clonazepam only) [75–80]. Nevertheless, benzodiazepines are not indicated for the long-term treatment of anxiety, insomnia, or stress, for which they are most often prescribed [81–83]. While benzodiazepines may be taken for a brief period of days or possibly a few weeks to manage stress or anxiety in a crisis or other short-term situation, they are not intended as nightly sleep aids or daily calming agents. While the chronic use of benzodiazepines is often contraindicated, benzodiazepines are sometimes prescribed for years. Good prescribing practice for benzodiazepines is to use the lowest effective dose for the shortest period of time, limiting prescription duration to no more than four to six weeks in most cases. Benzodiazepine therapy should never be initiated without patient education about the drug's proper use and the risk of possible long-term complications and an exit plan clearly defined by the prescriber. Even short-term use of benzodiazepines may be problematic, because physiologic dependence on benzodiazepines can occur after two to four weeks of use [84–89].

The acute withdrawal syndrome, with which clinicians are acquainted, is associated with the removal of the central nervous system depressant drugs from brain GABA receptors they occupied during benzodiazepine treatment and the associated autonomic hyperactivity that ensues. Acute withdrawal follows a short and relatively predictable trajectory; the literature provides guidelines to manage this syndrome, including tapering protocols [72,90]. Protracted symptoms,

formerly called post-acute withdrawal syndrome but likely better described as BIND, occur long after benzodiazepines are eliminated from the body. This suggests another mechanism, one other than withdrawal, is in play; this mechanism is likely the lasting neurotoxic consequences of benzodiazepines. Neurotoxicity is a complex condition which may or may not be reversible and which can manifest differently in different patients [91–93]. For those individuals who experience debilitating symptoms long after complete benzodiazepine cessation, the concept of BIND not only explains these symptoms, but may help such patients find support in the healthcare system. However, the neurobiological underpinnings of these phenomena need to be established before they can be fully accepted by the scientific community. Further study is needed. A limitation of this work was that no randomized clinical trials made as an endpoint enduring symptoms following complete benzodiazepine cessation, so the protracted symptoms that were observed must be considered signals.

Benzodiazepines are powerful agents, the use of which may lead to long-term neurologic dysfunction in some patients. The patient's susceptibility, the extent, and reversibility of these neurologic changes is not known. There is an urgent need to answer those questions and limit long-term benzodiazepine use in accordance with the Hippocratic oath: *first, do no harm*.

While BIND is a newly evolving concept and lacks diagnostic criteria, broad medical acceptance, and treatment protocols, this scoping review provides evidence that these prolonged symptoms are not outliers, but valid scientific observations that merit greater study.

## Acknowledgments

The authors of this paper must posthumously acknowledge a great debt they owe to Christy Huff, MD, who passed away in March 2024 before this scoping review could be completed. Huff was a core member of a group of collaborators who worked on three benzodiazepine papers based on an online survey that she herself helped to compose and administer. She was zealous in educational efforts for the benzodiazepine community, and she was among the first authors to conceive and help formulate the protocol for this scoping review. Her work on behalf of benzodiazepine safety is well known, deserves praise, and will benefit many patients in years to come. She was an uncompromising patient advocate, but, to us, she was also a dear friend and beloved colleague. The authors regret that she cannot be recognized as an author on this paper, as she has left us before it could be completed, but all of the authors gratefully acknowledge her work, her support, her intelligence, and the fact that none of this would have been possible without her. The authors would like to acknowledge and thank the following contributors. Jaden Brandt (BSc.Pharm, MSc., DipPH) contributed to protocol development, drafting of abstract, and initial screening of articles. Jaden is a clinical instructor at University of Manitoba. Anthony Cao assisted in the abstract draft process as well as initial screening process. He is a student at St. Joseph's University in his final year of studies. The authors also acknowledge the contribution of Jo Ann LeQuang of Angleton, Texas, a medical writer, whose services were covered by the Alliance for Benzodiazepine Best Practices.

## Author contributions

**Conceptualization:** Alexis D. Ritvo, Bernard Silvernail.

**Data curation:** Christi Piper.

**Investigation:** Kyla N. Shade, Alexis D. Ritvo, Bernard Silvernail, A. J. Reid Finlayson, Jolene E. Bressi, D. E. Foster, Ian J. Martin, Peter R. Martin.

**Methodology:** Kyla N. Shade, Alexis D. Ritvo, Christi Piper.

**Project administration:** Kyla N. Shade, Alexis D. Ritvo, Bernard Silvernail.

**Supervision:** Bernard Silvernail.

**Writing – original draft:** Kyla N. Shade, Alexis D. Ritvo, Bernard Silvernail, A. J. Reid Finlayson, Jolene E. Bressi, D. E. Foster, Ian J. Martin, Peter R. Martin.

**Writing – review & editing:** Kyla N. Shade, Alexis D. Ritvo, Bernard Silvernail, A. J. Reid Finlayson, Jolene E. Bressi, D. E. Foster, Ian J. Martin, Peter R. Martin.

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
