## [Decision Letter · Decision Letter 0]

5 Jun 2025

Dear Dr. Ritvo,

Thank you for submitting your manuscript to PLOS ONE. After careful consideration, we feel that it has merit but does not fully meet PLOS ONE’s publication criteria as it currently stands. Therefore, we invite you to submit a revised version of the manuscript that addresses the points raised during the review process.

We look forward to receiving your revised manuscript.

Kind regards,

Hira Rafi

Academic Editor

PLOS ONE

Journal Requirements:

[Alexis Ritvo is contracted as the medical director for the national non-profit the Alliance for Benzodiazepine Best Practices (501(c)(3)). Bernie Silvernail is the CEO of the Alliance for Benzodiazepine Best Practices.].

If your submission does not contain these data, please either upload them as Supporting Information files or deposit them to a stable, public repository and provide us with the relevant URLs, DOIs, or accession numbers. For a list of recommended repositories, please see https://journals.plos.org/plosone/s/recommended-repositories

Reviewers' comments:

Reviewer's Responses to Questions

**Comments to the Author**

1. Is the manuscript technically sound, and do the data support the conclusions?

Reviewer #1: Partly

2. Has the statistical analysis been performed appropriately and rigorously?

Reviewer #1: N/A

3. Have the authors made all data underlying the findings in their manuscript fully available?

Reviewer #1: No

4. Is the manuscript presented in an intelligible fashion and written in standard English?

Reviewer #1: Yes

Reviewer #1: By Kyla N. Shade, Alexis D. Ritvo, Bernard Silvernail, A. J. Reid Finlayson, Jolene E. Bressi,　D.E. Foster, Ian J. Martin, Christi Piper, and Peter R. Martin reported a research article entitled, “Long-Term Neurological Consequences Following Benzodiazepine Exposure: A Scoping Review” to PLOS ONE.

The authors should clarify the rationale for defining a two- to four-week period as the threshold for benzodiazepine exposure risk.

As various benzodiazepines were included, and the reviewed studies do not consistently examine the same agents, the authors are encouraged to discuss both the differences and any potential uniformities across these drugs.

Furthermore, the duration of use, the ratio of duration to discontinuation, and the timing of discontinuation should be more clearly addressed in relation to the development of withdrawal symptoms.

**Do you want your identity to be public for this peer review?** For information about this choice, including consent withdrawal, please see our Privacy Policy

Reviewer #1: No

---

## [Author Response · Author response to Decision Letter 1]

28 Jul 2025

Dear Editors:

Thank you for the peer review comments which we have addressed below. This Response to Reviewers document contains all three responses to reviewers. Please also see the Response to Reviewers letter where these responses are delineated in a table.

1 Is the manuscript technically sound, and do the data support the conclusions?

Comments: Partly

Response: This is a scoping review and meets the standards for such. Without more specifics, it is difficult for us to address

this.

2 Has the statistical analysis been performed appropriately and rigorously?

Comments: Agreed

Response: None

3 Have the authors made all data underlying the findings in their manuscript fully available?

Comments: No

Response: This comment confuses us. We have supplied a detailed search strategy (databases, flowchart, keyword lists) and inclusion/exclusion criteria. We feel that our research could be duplicated by any who followed the methodology we disclosed. We do not know what other underlying data are required

4. Is the manuscript presented in an intelligible fashion and written in standard English?

Comment: Yes

Response: Thank you

5A. Reviewer comments invited: The authors should clarify the rationale for defining a two- to four-week period as the threshold for benzodiazepine exposure risk.

Response: We established the study criteria as patients ≥18 years who had taken BZD chronically, long terms, or consistently over a period of > 2 wk and who had discontinued them for >4 weeks. This is in line with the published literature and the literature review recently conducted for the ASAM BZD tapering guidelines. This guidance states that physical dependence may occur in as little as 2 wk and symptoms of protracted withdrawal may emerge as early as 4 wk post-discontinuation. We originally cited the FDA boxed warning for this (original reference #84). We substituted the ASAM guidelines (Brunner 2025), which makes the statements noted in the column to the left and also references the FDA boxed warning. We identified 8 additional references used to support this statement, and have included only the 5 with the highest quality evidence. If more references are desired, we will gladly add the other three: Maine BZD study group 2008, Pottie 2018, and Hood 2014.

5B. As various benzodiazepines were included, and the reviewed studies do not consistently examine the same agents, the authors are encouraged to discuss both the differences and any potential uniformities across these drugs.

Response: This was a scoping review looking for evidence of protracted withdrawal within a class of medications. This is a common approach in this sort of research. A good example of this approach is NIH Project Number5R01AG060975 which treats benzodiazepines as a class.

Since this is the first such study of its kind, the authors felt that it was wise to approach it in broad strokes. Fine-tuning it to specific agents exceeds our original goals. This would be a worthwhile topic for future research or a secondary analysis. However, as we started the project, it was unclear what, if anything, we would find.

5C. Furthermore, the duration of use, the ratio of duration to discontinuation, and the timing of discontinuation should be more clearly addressed in relation to the development of withdrawal symptoms.

Response: The withdrawal symptoms in our study are not described by the studies in any in-depth way given the broad nature of studies accepted by our scoping review. We reported mainly on signals. The data as to duration of use, etc. to which we could reliably anchor these findings is spotty and inconsistent between the studies. None of the studies were looking for withdrawal symptoms but mentioned them as incidental findings.

The interest in associating duration of use, etc. to our findings might introduce what could be potential bias to our scoping review. Our purpose was to look for the presence of withdrawal symptoms and not adjudicate their causes.

By only looking for specific things (e.g., withdrawal related to duration of use), we defeat the purpose of the scoping review, which was just to look for evidence for withdrawal symptoms after benzodiazepine discontinuation.

6. PLOS authors have the option to publish the peer review history of their article. If published, this will include your full peer review and any attached files.

Response: None

In conclusion, we appreciate the comments offered to us, in particular suggestions that would be very worthwhile subjects for secondary analysis or future endeavors. Our scoping review is a first step in these efforts, and we hope not the last in our investigation into this subject.

We acknowledge the previously noted competing interests. This does not alter our adherence to PLOS ONE policies on sharing data and materials.

Note that the "Combined Response to Reviewers" document filed under the Response to Reviewers category also contains our response to the PLoS One request for information to ensure the replicability of the study, which was emailed to Alexis Ritvo by Johnelle Ryan Razo on 6/22/2025.

---

## [Editor Report · Decision Letter 1]

30 Jul 2025

Long-term neurological consequences following benzodiazepine exposure: A scoping review.

PONE-D-25-04510R1

Dear Dr. Ritvo,

We’re pleased to inform you that your manuscript has been judged scientifically suitable for publication and will be formally accepted for publication once it meets all outstanding technical requirements.

Kind regards,

Hira Rafi

Academic Editor

PLOS ONE
---

## [Editor Report · Acceptance letter]

PONE-D-25-04510R1

PLOS ONE

Dear Dr. Ritvo,

I'm pleased to inform you that your manuscript has been deemed suitable for publication in PLOS ONE. Congratulations! Your manuscript is now being handed over to our production team.

Kind regards,

on behalf of

Dr. Hira Rafi

Academic Editor

PLOS ONE